# Multi-Model Strategies for Prevention of Infection Caused by Certain Multi-Drug Resistant Organisms in A Rehabilitation Unit: A Semi-Experimental Study

**DOI:** 10.3390/antibiotics12071199

**Published:** 2023-07-18

**Authors:** Shiyu Li, Ji Lin, Siyuan Tao, Linwen Guo, Wenzhi Huang, Jingwen Li, Chunping Du, Zhiting Wang, Liwen Liu, Yi Chen, Fu Qiao

**Affiliations:** 1Department of Rehabilitation Medicine, West China Hospital, Sichuan University, Chengdu 610041, China; lishiyu@wchscu.cn (S.L.); duchunping118@wchscu.cn (C.D.); 2Department of Infection Control, West China Hospital, Sichuan University, Chengdu 610041, China; linji@scu.edu.cn (J.L.); taosiyuan@wchscu.cn (S.T.); guolw@scu.edu.cn (L.G.); huangwenzhi@wchscu.cn (W.H.); lijingwen@wchscu.cn (J.L.); 3Department of Operation Management, West China Hospital, Sichuan University, Chengdu 610041, China; wangzhiting@wchscu.cn; 4Department of Equipment and Materials, West China Hospital, Sichuan University, Chengdu 610041, China; liuliwen@scu.edu.cn

**Keywords:** multi-model strategies, multi-drug resistant organisms, healthcare-associated infections, rehabilitation unit

## Abstract

**Objective:** To assess the effectiveness of multi-model strategies on healthcare-associated infections (HAIs) caused by multi-drug resistant organisms (MDROs) in rehabilitation units. **Methods:** A semi-experimental study was conducted in a rehabilitation unit with 181 beds from January 2021 to December 2022 in a teaching hospital with 4300 beds in China. In 2021, many basic prevention and control measures were conducted routinely. Based on the basic measures, strengthening multi-model strategies for the prevention and control of MDROs was pursued year-round since 1 January 2022. **Results:** A total of 6206 patients were enrolled during the study period. The incidence density of HAIs caused by MDROs decreased from 1.22 (95% CI, 0.96~1.54) cases/1000 patient-days in the pre-intervention period to 0.70 (95% CI, 0.50~0.95) cases/1000 patient-days (*p* = 0.004). Similarly, the incidence of HAIs in the intervention period was 50.85% lower than that in the pre-intervention period (2.02 (95% CI, 1.50~2.72) vs. 4.11 (95% CI, 3.45–4.85) cases/100 patients, *p* < 0.001). The rate of MDROs isolated from the environment decreased by 30.00%, although the difference was not statistically significant (*p* = 0.259). **Conclusion:** Multi-model strategies can reduce the incidence of HAIs and HAIs caused by certain MDROs in the rehabilitation unit.

## 1. Introduction

In recent years, healthcare-associated infections (HAIs) caused by multi-drug resistant organisms (MDROs) have become severe challenges to clinical treatment, often prolonging the length of stay (LOS), increasing mortality, and increasing medical costs [1,2,3]. The prevalence of MDROs poses a threat to public health worldwide. A report published by the World Health Organization (WHO) estimates that there are more than 25,000 and 23,000 MDRO infection-related deaths every year in Europe and the United States, respectively. The annual economic burden caused by the infection of MDROs is EUR 1.5 billion and USD 3.4 billion in Europe and the United States, respectively [4]. Several recent studies also confirmed that the medical cost of patients with MDRO infections was 1.4~1.7 times that of patients without [5,6,7]. In China, one study conducted in 68 hospitals showed that the average increase in LOS due to infections caused by studied MDROs was 14 days [8].

In recent decades, with the development of early rehabilitation, many surgical patients are transferred to the rehabilitation unit within one or two days after surgery, and some critical patients are transferred to the rehabilitation unit directly from the intensive care unit (ICU). Moreover, many rehabilitation instruments are shared between different patients, and it is almost impossible to disinfect the instruments in time after each use [9]. All of these factors increase the risk of HAIs for patients hospitalized in rehabilitation units, especially the infection of MDROs. A study revealed that the MDRO test-positive rate was 16.70% (96/575) in neurorehabilitation ward patients [10]. Patients colonized or infected with MDROs may severely or moderately limit their rehabilitation outcome [5], and the infection during rehabilitation will increase the length of hospitalization, reduce the rehabilitative outcomes and increase the mortality rate significantly [7]. A bundle of interventions, including education and training, hand hygiene promotion, contact precaution, and environmental disinfection, have been confirmed to reduce the incidence of HAIs caused by MDROs [11,12,13,14]. However, there is less evidence on the strategies for MDRO prevention and control in rehabilitation units. To assess the effectiveness of multi-model strategies on infections caused by MDROs in the rehabilitation unit, especially in carbapenem-resistant *Acinetobacter baumannii* (CRAB) and carbapenem-resistant *Enterobacteriaceae* (CRE) endemic areas, we, therefore, conducted this study.

## 2. Results

### 2.1. Characteristics of Patients in The Pre-Intervention and Intervention Periods

A total of 7200 patients were admitted to the rehabilitation unit from January 2021 to December 2022, and 6206 patients of them were eligible (Figure 1). A total of 3262 patients and 2944 patients were included in the pre-intervention and intervention periods, respectively (S). The mean age was 53.87 ± 16.13 years in the intervention period, which was significantly higher than that in the pre-intervention period (*p* < 0.001). Most of the underlying diseases were no significant difference between the two groups, although the proportion of patients with respiratory failure and malignant tumors was statistically different between the two groups, the intervention period was higher than that in the pre-intervention period, 2.65% vs. 1.69% (*p* = 0.009) and 6.69% vs. 4.20% (*p* < 0.001), respectively. A total of 19.13% (1187/6206) of the patients had a history of surgery during their hospitalization, but there was no significant difference between the two groups. Forty-five patients had at least one MDRO infection at admission in the intervention period, which was slightly higher than the pre-intervention group, but there was no significant difference (*p* = 0.552). The details of the two groups are shown in Table 1.

### 2.2. The Compliance of Infection Control Measures

The rate of family caregivers for patients decreased to 38.48% in the intervention period. Hand hygiene compliance during the intervention period was 89.08% (1771/1988) while it was 90.54% in the pre-intervention period (*p* = 0.071). The differences in the consumption of liquid soap and alcohol-based hand rub between the intervention and pre-intervention periods were also not statistically significant (S). The consumption of gloves in the intervention period was 2.15 ± 0.26 pairs per patient day which was statistics significantly higher than that in the pre-intervention period (*p* = 0.010), while the consumption of gowns was 64.67 ± 10.01 pieces per 1000 patient days, which was higher than that in the pre-intervention period but not statistical significance (*p* = 0.892). 

At the end of December 2021, 84 environmental samples, including bed rails and tables, were collected randomly from the wards, and 18 (21.34%) of them were positive for MDROs, including 14 CRKP and 4 CRAB strains. In March 2023, environmental sampling was conducted again, and 100 samples were taken randomly from the wards. Of the 100 samples, 15 (15.00%) were positive for MDROs, including 10 CRKP strains and 5 CRAB strains. No other target MDRO was isolated from the environmental sampling. The environmental pollution rate of MDROs decreased by 30.00%, although the difference was not statistical significance (*p* = 0.259). The details are shown in Table 2.

### 2.3. Healthcare-Associated Infections Caused by MDROs

In total, 134 cases and 60 cases of HAIs were confirmed in the pre-intervention period and intervention period, respectively, and the incidence of HAIs was significantly lower in the intervention period than in the pre-intervention period (2.04% (95% CI, 1.56~2.62) vs. 4.11% (95% CI, 3.45~4.85), *p* < 0.001). In those infection cases, 113 cases of MDROs were isolated from 109 patients more than 48 h after admission during the study period, and the total incidence density of HAIs caused by MDROs was 0.97 (95% CI, 0.96–1.54) cases per 1000 patient days. CRKP was the most common pathogen, accounting for 35.40% of HAIs caused by MDROs. Lower respiratory tract infections were the most common sites of HAIs, followed by urinary tract infections, which accounted for 62.83% (71/113) and 30.09% (34/113) of the whole infection site, respectively.

During the intervention period, only 39 cases of MDROs were isolated from eligible patients, and the incidence density of HAIs caused by MDROs was significantly decreased compared with that during the pre-intervention period (0.70 (95% CI, 0.50–0.95) cases per 1000 patient-days vs. 1.22 (95% CI, 0.96–1.54) cases per 1000 patient days, *p* = 0.004). HAIs caused by CRKP decreased from 0.48 (95% CI, 0.32–0.69) cases per 1000 patient days to 0.20 (95% CI, 0.10–0.35) cases per 1000 patient days, *p* = 0.008. However, HAIs caused by CRAB decreased by 47.37%, but the difference was not statistically significant (0.20 (95% CI, 0.10–0.35) cases per 1000 patient days vs. 0.38 (95% CI, 0.24–0.57) cases per 1000 patient days). The incidence density of HAIs caused by other MDROs was also not significantly decreased between the intervention period and pre-intervention period because of low incidence density. The monthly variation of the incidence density is shown in Figure 2.

In terms of infection site, both the incidence density of lower respiratory tract infection and urinary tract infection caused by MDROs were decreased statistically significantly between the intervention period and pre-intervention period, that is, 0.43 (95% CI, 0.27–0.64) cases per 1000 patient-days vs. 0.78 (95% CI, 0.57–1.03) cases per 1000 patient days and 0.18 (95% CI, 0.09–0.33) cases per 1000 patient days vs. 0.40 (95% CI, 0.25–0.59) cases per 1000 patient days, separately. The details of the incidence density between the intervention period and the pre-intervention period are shown in Table 3. 

## 3. Materials and Methods

### 3.1. Study Design 

A semi-experimental study was conducted in a rehabilitation unit with 181 beds from 1 January 2021 to 31 December 2022 in a teaching hospital with 4300 beds in mainland China. This unit was major for spinal cord injury disease, musculoskeletal disease, and nervous system disease. In 2021, a bundle of basic prevention and control measures was conducted routinely. Based on the basic measures, strengthening multi-model strategies for the prevention and control of infections caused by MDROs have been implemented all year round since 1 January 2022, which was the intervention period.

### 3.2. Inclusion and Exclusion Criteria

All the patients (Age ≥18 years) older than 18 years and above admitted to the rehabilitation unit in the study period were included. Patients were excluded if they were (1) Age < 18 years; (2) stayed in the unit for less than 48 h; or (3) had a length of stay between 2021 and 2022. 

### 3.3. Prevention and Control Measures in The Pre-Intervention Period

(1) Notification. A notification phone call was given to the doctors in charge by microbiology laboratory workers immediately when the MDRO was isolated, and the results were uploaded to the laboratory information system (LIS) immediately. A contact precaution marker was labeled automatically on the hospital information system (HIS). (2) Contact precautions. Patients with MDROs were isolated when the nurse received the notification, and a contact precaution label was posted on the patient’s bed. All healthcare workers must wear gloves and gowns during care work. The infection control link nurses in the ward checked the compliance of contact precautions, and the infection control practitioners did an audit within 24 h to ensure that the clinicians correctly understood the communication and, therefore, consequently acted. (3) Reduce the number of people. Only three doctors were allowed to take the turnaround at the same time. One fixed family member or one professional nurse assistant was permitted to accompany, and only one visitor was enabled each time. (4) Routine disinfection. All of the rehabilitation instruments were cleaned and disinfected with 500 mg/L effective chlorine disinfectant twice a day.

### 3.4. Multi-Model Strategies for Prevention of HAIs Caused by MDROs in The Intervention Period

#### 3.4.1. System Change

An MDRO prevention and control work group was established in the unit, which was composed of a vice director, chief rehabilitation technician, head nurse, medical team leader, nursing team leader, infection control link nurse, and infection control professional. The responsibilities of the group included (1) coordinating and allocating beds in the whole unit to cohort patients with MDROs; (2) analyzing and discussing the reason for infection case by case; and (3) drafting prevention and control guidelines.

#### 3.4.2. Personnel and Behavior Management

(1) Family visits were canceled during hospitalization. (2) Whether the patients needed to be cared for by family members was assessed by the doctor in charge based on the patient’s condition. None of the family members were permitted to care for patients in the quasi-intensive care unit. (3) One doctor and one nurse were fixed in the quasi-intensive care unit to conduct the ward rounds and care work per shift. The treatment and ward rounds were carried out at different intervals. Only one technician was allowed to work at the same time except in emergencies. 

#### 3.4.3. Education and Training 

(1) All of the new healthcare workers must finish the course of prevention and control measures for MDROs before they engage in clinical work. (2) Regular education was carried out for different types of professions. (3) A training and test of infection prevention and control measures must be completed for doctors and nurses before they work in a quasi-intensive care unit. 

#### 3.4.4. Communication and Data Feedback

(1) Weekly communication meetings were built between infection control practitioners and link nurses to feedback on the audit results and discuss work plans for next step since January 2022. (2) All of MDROs data in the pre-intervention period were feedback to the management group in January 2022. Then, a quarterly data feedback meeting was conducted with the directors, head nurse, and other management group members. (3) Statistical data of HAIs of MDROs was feedback to all of the healthcare workers on morning shift irregularly. 

#### 3.4.5. Environmental Control

(1) All instruments and equipment were disinfected by the rehabilitation therapist immediately after use by the MDRO’s patient. (2) The concentration of disinfectant was checked by a designated nurse every day. (3) A fluorescent marker was used to check compliance with cleaning twice a week. (4) Environmental surveillance culture was conducted to assess the environmental burden of MDROs. The environment swabbing was taken by sterile rayon swabs (Copan, Brescia, Italy) moistened with tryptic soy broth (TSB, Hopebio, Qingdao, China). The swabs were immediately placed into 15 mL sterile tubes containing 6 mL of TSB. The tubes were incubated at 37 °C overnight and centrifuged. The supernatant was discarded, and precipitates were resuspended in 1 mL of TSB. A 50 µL suspension was streaked onto Acinetobacter selected-agar plates (CHROMagarTM, Paris, France) containing 4 μg/mL meropenem to screen CRAB, Pseudomonas selected-agar plates (CHROMagarTM, Paris, France) containing 4 μg/mL meropenem to screen CRPA, Simmon citrate agar plates containing 2 μg/mL meropenem to screen CRK, Orientation selected-agar plates (CHROMagarTM, Paris, France) containing 2 μg/mL meropenem to screen CRE, MRSA, and VRE selected-agar plates (CHROMagarTM, Paris, France) to screen MRSA and VRE separately. The plates were incubated at 37 °C for 18~24 h, and suspected colonies were subjected to preliminary species identification based on matrix-assisted laser desorption/ionization–time of flight mass spectrometry (Bruker, Billerica, MA, USA).

### 3.5. Definitions

The diagnostic criteria for HAIs were the diagnostic standards for HAIs published by the Ministry of Health in 2001 [15]. The certain MDROs in this study included Carbapenem-resistant *Acinetobacter baumannii* (CRAB), Carbapenem-resistant *Enterobaceteriacae* (CRE), Carbapenem-resistant *Pseudomonas aeruginosa* (CRPA), Vancomycin-resistant *Enterococci* (VRE), and Methicillin-resistant *Staphylococcus aureus* (MRSA). 

### 3.6. Main Outcomes

The primary study outcome was the incidence density of HAIs caused by MDROs in inpatients. The secondary outcome was the incidence density of HAIs. We also report the compliance of hand hygiene and usage of protective equipment during two periods.

### 3.7. Clinical Data Collection and Statistical Analysis

Demographic and clinical data of eligible patients were collected, including gender, age, diagnosis, underlying disease, infection of MDROs at admission, and the length of stay.

For descriptive analysis, qualitative data were expressed in terms of frequency, and the incidence of MDROs was expressed as incidence density (cases per 1000 patient days). Quantitative variables with a normal distribution were expressed as the mean ± standard deviation (SD), which were expressed as median (25% percentile, 75% percentile) if the normal distribution was not met. Qualitative data were analyzed by the Chi-square test or Fisher’s exact test, Kolmogorov–Smirnov normality test was used for normality test. *T*-test was used if quantitative variables were normal distribution; otherwise, Mann–Whitney U-test was conducted. The difference of MDROs incidence density among interventions was analyzed by Poisson analysis, and RR was used to indicate the relative risk. Analyses were performed using R v. 4.3.0.

## 4. Discussion

Patients colonized or infected with MDROs may affect their rehabilitation programs. One survey conducted in Germany showed that 27% of rehabilitation facilities refused to accept patients with MDROs, and only 27% of the rehabilitation centers allowed patients with MDROs to participate in full rehabilitation programs [16]. Another survey conducted in Europe showed that patients with MDROs wait longer for admission in 36% of facilities and have been refused admission in 11% of facilities [17]. However, the prevalence of MDROs in rehabilitation units is still very high and sometimes causes outbreaks [9]. A multicenter observation study for neurorehabilitation showed that 55% of HAIs needed functional isolation due to multidrug-resistant germs [18]. Another study in Germany showed that 2.2% of general rehabilitation patients were colonized with VRE, and 6.7% of them had multi-drug-resistant gram-negative pathogens [19]. In this study, we found out that the incidence of HAIs caused by MDROs in the pre-intervention period was 2.27% (1.22 cases per 1000 patient days), and the most common MDROs were CRKP, CRAB, and CRPA. The incidence is lower than that in a previous study conducted in Southwest China and some other studies [10], which may be because of the difference in case definition. However, the concept of MDROs was similar. 

How patients with MDRO infections should be managed in a rehabilitation setting is still lacking due to a survey in Europe [17]. Effective strategies to prevent HAIs caused by MDROs are still limited. In our study, multi-model strategies, including system change, personnel and behavior management, education and training, communication and data feedback, and environmental control, were carried out in the intervention period in the rehabilitation unit, and the incidence of HAIs caused by MDROs decreased significantly through the strategies during the intervention period, even though the infection rate at admission was slightly higher. We indeed explored a way to decrease HAIs caused by MDROs in the rehabilitation department. System change may play a key role in our multi-strategies, and one study conducted in the China Rehabilitation Research Centre decreased HAIs caused by MDROs through the PDCA cycle. In their study, the detection rate of MRSA and CRPA decreased significantly, but that of MDRAB did not [20]. 

In this study, the average age in the pre-intervention period was lower than that in the intervention period, but the difference was only 1.73 years. Several studies reported that age was an independent risk factor for MDROs infection [21,22]. The older the age, the higher the risk of infection. Although the average age in the intervention group was higher than that in the control group in this study, there was still a decreased incidence of MDROs, indicating a relatively robust result. Respiratory failure, which reflected the severity of the patients’ illness, has been reported as an independent risk factor for MDROs infection [23]. In this study, the proportion of respiratory failure in the intervention period was higher than that in the pre-intervention period; there was still a decrease in the infection rate of MDROs, which indicated a relatively robust result.

Personal and behavior management are always hard to execute. In China, the rate of family caregivers is as high as 90% for cultural reasons [24], and compliance with hand hygiene and other infection control measures is very low, even when the patients they care for have communicable diseases [25,26]. In this study, to decrease the rate of family caregivers and reduce the number of persons in each ward, family visits were canceled, and each family caregiver was evaluated by the doctor in charge. The family caregiver rate decreased to 38.48% in 2022. However, we did not collect the details of family caregivers all year round in 2021, and the exact rate of family caregivers was not acquired, but we estimate that the rate was as high as 65% based on limited data collected by the head nurse.

Compliance with environmental cleaning for the immediate surrounding area is also a key recommendation to decrease infection caused by MDROs [27]. One study showed that the MDRO-positive rate was 7.7% in the common area and rehabilitation gym environment [28]. In this study, the contamination rate of common areas in the rehabilitation unit was 21.43%, which is much higher than that in a previous study. However, the contamination rate decreased by 30% through the intervention. This may contribute to reducing the risk of MDRO transmission. Although the consumption of gloves increased during the intervention period, there might have been misused gloves, so the consumption of gloves only partially reflects compliance.

There are several limitations to this study. First, the study is a single-center study in an area where MDROs are notably prevalent, especially CRAB and CRKP, which limits the extrapolation of the results to primary hospitals; however, it has significance for areas with high MDRO prevalence. Second, this study was conducted in the rehabilitation unit that received many severe patients transferred from ICUs or surgical units in a large teaching hospital, which also limits the reference to other units and other rehabilitation units receiving traditional patients, but it has significance for the rehabilitation units that are developing rapid rehabilitation medicine. Finally, we did not conduct active surveillance screening of MDROs for the patients on admission and during their hospitalization, and we could not evaluate the effect of multi-strategies on colonized organisms. 

## 5. Conclusions

This semi-experimental study found that the comprehensive multi-model strategies reduced the incidence of HAIs and the HAIs caused by MDROs. It also reduced the contamination rate of MDROs in patients’ room environments. These findings demonstrate that these interventions can effectively decrease the burden of MDROs in rehabilitation units.

## Figures and Tables

**Figure 1 antibiotics-12-01199-f001:**
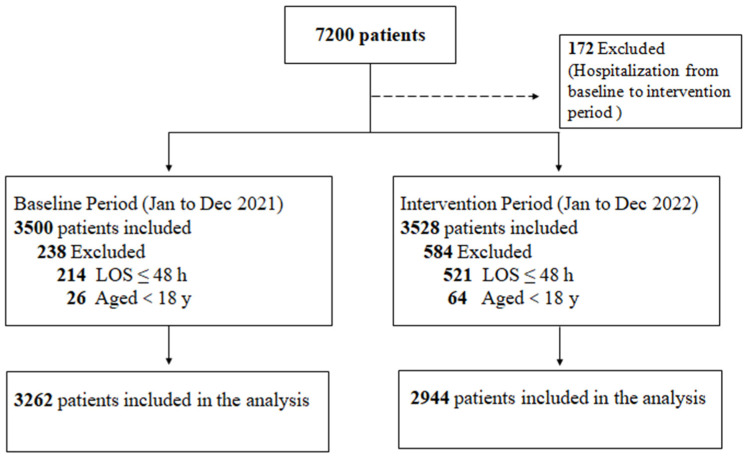
Flow chart and cohorts for analyses.

**Figure 2 antibiotics-12-01199-f002:**
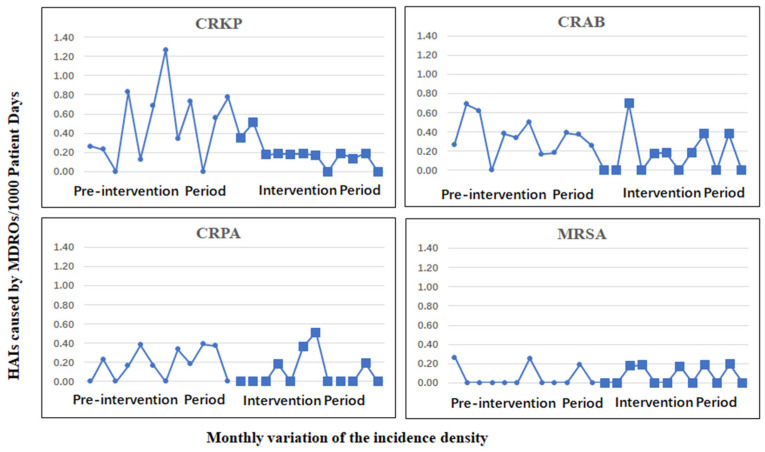
Monthly incidence density of HAIs caused by CRKP, CRAB, CRPA, and MRSA. Abbreviations: HAIs, healthcare-associated infections; CRKP, carbapenem-resistant *Klebsiella pneumoniae*; CRAB, carbapenem-resistant *Acinetobacter baumannii*; CRPA, carbapenem-resistant *Pseudomonas aeruginosa*; MRSA, methicillin-resistant *Staphylococcus aureus*.

**Table 1 antibiotics-12-01199-t001:** Characteristics of the study population in each period.

	All Patients (*n* = 6206)	Pre-Intervention Period (*n* = 3262)	Intervention Period (*n* = 2944)	*p* Value
Age, mean (SD), y	52.96 (16.75)	52.14 (17.25)	53.87 (16.13)	<0.001
Sex (*n*, %)				
Female	2646 (42.63)	1313 (40.25)	1333 (45.28)	<0.001
Male	3560 (57.36)	1949 (59.75)	1611 (54.72)	
Underlying diseases (*n*, %)				
Hypertension	1797 (28.96)	959 (29.40)	838 (28.46)	0.418
Diabetes	770 (12.41)	404 (12.39)	366 (12.43)	0.955
Respiratory failure	133 (2.14)	55 (1.69)	78 (2.65)	0.009
Tuberculosis	40 (0.64)	24 (0.74)	16 (0.54)	0.345
Heart failure	103 (1.66)	46 (1.41)	57 (1.94)	0.105
Renal failure	93 (1.5)	50 (1.53)	43 (1.46)	0.815
Hepatic insufficiency	82 (1.32)	40 (1.23)	42 (1.43)	0.490
Malignant tumors	334 (5.38)	137 (4.20)	197 (6.69)	<0.001
Hematological diseases	5 (0.08)	2 (0.06)	3 (0.10)	0.909
COPD	59 (0.95)	27 (0.83)	32 (1.09)	0.293
HIV	5 (0.08)	2 (0.06)	3 (0.1)	0.909
Hemiplegia/Paraplegia	1306 (21.04)	687 (21.06)	619 (21.03)	0.973
Mortality (*n*, %)	15 (0.24)	9 (0.28)	6 (0.20)	0.564
Surgery (*n*, %)	1187 (19.13)	600 (18.39)	587 (19.94)	0.065
LOS, median (IQR), d	19 (12–22)	20 (13–22)	18 (11–23)	0.467
MDROs at admission (*n*, %)	89 (1.43)	44 (1.35)	45 (1.53)	0.552

Abbreviations: COPD, chronic obstructive pulmonary disease; HIV, human immunodeficiency virus; LOS, length of stay.

**Table 2 antibiotics-12-01199-t002:** The process index in each period.

Process Indicators	Pre-Intervention Period	Intervention Period	*p* Value
Compliance with hand hygiene (*n*, %)	3937 (90.54%)	1771 (89.08%)	0.071
HH equipment consumption (mL per patient day)	27.66	27.38	0.923
Liquid soap (mL per patient day)	7.84 ± 3.18	10.92 ± 6.53	0.156
ABHR (mL per patient day)	19.83 ± 5.36	16.46 ± 4.67	0.115
PPE usage			
Gloves (pairs per patient day)	1.42 ± 0.11	2.15 ± 0.26	0.010
Gowns (pieces per 1000 patient day)	60.75 ± 14.72	64.67 ± 10.01	0.892
Environment surveillance culture (*n*, %)	18 (21.43%)	15 (15.00%)	0.259
CRKP	14 (16.67%)	10 (10.00%)	0.182
CRAB	4 (4.76%)	5 (5.00%)	0.940

Abbreviations: HH, hand hygiene; ABHR, alcohol-based hand rub; PPE, personal protective equipment; CRKP, carbapenem-resistant *Klebsiella pneumoniae*; CRAB, carbapenem-resistant *Acinetobacter baumannii*.

**Table 3 antibiotics-12-01199-t003:** Effect of multi-strategies on HAIs caused by MDROs.

	Pre-Intervention Period		Intervention Period		*RR* (95% CI)	*p* Value
	Case (*n*)	Incidence Density ^a^ (95% CI)	Case (*n*)	Incidence Density ^a^ (95% CI)		
MDROs	74	1.22 (0.96, 1.54)	39	0.70 (0.50, 0.95)	0.57 (0.39, 0.84)	0.004
CRE	34	0.56 (0.39, 0.79)	16	0.29 (0.16, 0.46)	0.51 (0.28,0.92)	0.022
CRKP	29	0.48 (0.32, 0.69)	11	0.20 (0.10, 0.35)	0.41 (0.20, 0.82)	0.008
CR-*E. cloacae*	4	0.07 (0.02, 0.17)	1	0.02 (0.01,0.10)	0.27 (0.30, 2.42)	0.199
CR-*K. oxytoca*	1	0.02 (0, 0.09)	2	0.04 (0.01, 0.13)	2.16 (0.19, 23.86)	0.262
CR-*E.coli*	0	0 (0, 0.06)	2	0.04 (0.01, 0.13)	--	--
CRAB	23	0.38 (0.24, 0.57)	11	0.20 (0.10, 0.35)	0.52 (0.25, 1.06)	0.064
CRPA	13	0.21 (0.11, 0.37)	7	0.13 (0.05, 0.26)	0.58 (0.23, 1.46)	0.249
MRSA	3	0.05 (0.01, 0.14)	5	0.09 (0.03, 0.21)	1.80 (0.43, 7.54)	0.209
VRE	1	0.02 (0.01, 0.09)	0	0 (0, 0.07)	--	--
MDRO Infection site	71	1.17 (0.92, 1.48)	34	0.61 (042, 0.85)	0.52 (0.34, 0.78)	
LRTI	47	0.78 (0.57, 1.03)	24	0.43 (0.27, 0.64)	0.55 (0.34, 0.90)	0.015
UTI	24	0.40 (0.25, 0.59)	10	0.18 (0.09, 0.33)	0.45 (0.22, 0.94)	0.027

Notes: ^a^ cases per 1000 patient days. Abbreviations: CI, confidence interval; *RR*, relative risk; MDROs, multi-drug resistant organisms; CRE, carbapenem-resistant *Enterobaceteriacae*; CRKP, carbapenem-resistant *Klebsiella pneumoniae*; CRAB, carbapenem-resistant *Acinetobacter baumannii*; CRPA, carbapenem-resistant *Pseudomonas aeruginosa*; VRE, vancomycin-resistant *Enterococci*; MRSA, methicillin-resistant *Staphylococcus aureus*; LRTI, lower respiratory tract infection; UTI, urinary tract infection.

## Data Availability

The data presented in this study are available on request from the corresponding author. The data are not publicly available due to privacy.

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
