# Peer review of "Multi-Model Strategies for Prevention of Infection Caused by Certain Multi-Drug Resistant Organisms in A Rehabilitation Unit: A Semi-Experimental Study"

_antibiotics, 2023, doi:10.3390/antibiotics12071199_

Round 1

Reviewer 1 Report

The authors describes a multi-model strategies for prevention ICA in ICU setting the approach is a  semi-experimental study. The Study is of interesting particularly because there are no similar evaluation in China.

Just some minor comments:

Line 78: What way did the laboratory use to communicate the results? I mean by phone, by email, by LIS? Did they have any evidences that the clinician correctly understood the communication and therefore consequently act?

Line 91 what did you mean when you wrote chief technician? Was He a member of the laboratory staff?

Line 123. Please detailed environmental surveillance culture (which type of samples were collected, which type of culture method, eg. by using chromogenic media or what else? How long were they maintained in incubation in which condition?

Line 178 the consumption of gloves does not represent a key performance indicators. I could misuse gloves! Please add a comment in the discussion. 

Which is the role of microbiologists in this study. Microbiological data are paramount in the infection control program and the same is for microbiology professionals

Author Response

Dear Reviewer:

Thank you for your great comments concerning our manuscript entitled “Multi-model Strategies for Prevention of Infection Caused by Certain Multi-drug Resistant Organisms in A Rehabilitation Unit: A semi-experimental Study” (ID 2477387). Those comments are all valuable and very helpful for improving our paper, as well as the important guiding significance to our research. We have studied the comments carefully and have made corrections that we hope will be met with approval. The red font has been used to highlight the changes to the manuscript. The responses to your comments are as follows:

Comment 1

Line 78: What way did the laboratory use to communicate the results? I mean by phone, by email, by LIS? Did they have any evidences that the clinician correctly understood the communication and therefore consequently act?

Our response:

The MDRO prevention and control policy was reviewed and implemented in the hospital since July 2016. According to the policy, microbiological technicians informed the doctor in charge by phone immediately while a target MDRO was isolated, the results were uploaded to the LIS, and a contact precaution marker was labeled automatically on the Hospital Information System (HIS). When the clinicians received the notification, contact precaution was conducted. The infection control link nurses in the ward would check, and the infection control practitioners would audit within 24 hours to ensure that the clinicians correctly understood the communication and therefore consequently acted.

We have added the details on lines 78-87.

Comment 2

Line 91: what did you mean when you wrote chief technician? Was He a member of the laboratory staff?

Our response:

The chief technician is a head technician in charge of rehabilitation, not a member of the laboratory staff. We have changed “Chief technician” to “Chief rehabilitation technician” on line 97.

Comment 3

Line 123. Please detailed environmental surveillance culture (which type of samples were collected, which type of culture method, eg. by using chromogenic media or what else? How long were they maintained in incubation in which condition?

Our response:

All instruments and equipment were sampled using sterile rayon swabs (Copan, Brescia, Italy) moistened with tryptic soy broth (TSB, Hopebio, Qingdao, China). The swabs were immediately placed into 15 mL sterile tubes containing 6 mL of TSB. The tubes were incubated at 37°C overnight and centrifuged. The supernatant was discarded, and precipitates were resuspended in 1 mL of TSB. A 50 µl suspension was streaked onto Acinetobacter selected-agar plates (CHROMagarTM, Paris, France) containing 4 μg/mL meropenem to screen CRAB, Pseudomonas selected-agar plates (CHROMagarTM, Paris, France) containing 4 μg/mL meropenem to screen CRPA, Simmon citrate agar plates containing 2 μg/mL meropenem to screen CRK, Orientation selected-agar plates (CHROMagarTM, Paris, France) containing 2 μg/mL meropenem to screen CRE, MRSA and VRE selected-agar plates (CHROMagarTM, Paris, France) to screen MRSA and VRE separately. The plates were incubated at 37°C for 18 h~24 h, and suspected colonies were subjected to preliminary species identification based on matrix-assisted laser desorption/ionization–time of flight mass spectrometry (Bruker, Billerica, Massachusetts).

We have added the details on lines 129-143.

Comment 4

Line 178: the consumption of gloves does not represent a key performance indicators. I could misuse gloves! Please add a comment in the discussion:"

Our response:

It is true that there may be misuses of gloves, and the consumption of gloves can only partially reflect the compliance. We have added a comment in the discussion on lines 307-309.

Comment 5

Which is the role of microbiologists in this study. Microbiological data are paramount in the infection control program and the same is for microbiology professionals.

Our response:

In this study, microbiologists were mainly responsible for three things: 1) Conduct the environmental surveillance cultures. 2) Inform doctors in charge by phone immediately while the MDRO was isolated. 3) Upload the test results to LIS.

Thank you for these valuable suggestions. We have also made modifications carefully, hoping to further guide our research in the future.

Reviewer 2 Report

The efficacy of infection control bundles has been demonstrated previously although the improvements tend to reduce over time. The paper examines a rehabilitation unit specifically. Team work, education, feedback, disinfection after use and cleaning audit were the interventions beyond the standard contact precautions, staff limiting and disinfection twice a day. However, there was no mention of antibiotic stewardship. The control group age was lower than in the intervention cohort but this was not explained. As respiratory failure was more common was this a result of Covid-19? The recorded level of hand hygiene compliance was very high and possibly the result of direct observation rather than the actual rate of hand hygiene. Environmental swabbing would be expected to reflect the current patient flora as most MDRO would not have prolonged survival in the environment (MRSA, VRE and CRAB are exceptions). Reduction of MDRO in rehabilitation is beneficial and allows patients to be transferred more easily. Care provided by family maybe without hand hygiene could have a major effect on results and screening for MDRO was not performed.

The findings are not novel and care bundles have been widely examined. The effects of single mode of intervention would be of more interest. More detail on attempts to control antimicrobial prescribing practice and the relationship between patient bacterial flora and that in and away from their immediate environment is needed.

Author Response

Dear Reviewer:

Thank you for your great comments concerning our manuscript entitled “Multi-model Strategies for Prevention of Infection Caused by Certain Multi-drug Resistant Organisms in A Rehabilitation Unit: A semi-experimental Study” (ID 2477387). Those comments are all valuable and very helpful for improving our paper, as well as the important guiding significance to our research. We have studied the comments carefully and have made corrections that we hope will be met with approval. The red font has been used to highlight the changes to the manuscript. The responses to your comments are as follows:

Comment 1

The efficacy of infection control bundles has been demonstrated previously although the improvements tend to reduce over time. The paper examines a rehabilitation unit specifically. Team work, education, feedback, disinfection after use and cleaning audit were the interventions beyond the standard contact precautions, staff limiting and disinfection twice a day. However, there was no mention of antibiotic stewardship.

Our response:

Antibiotic stewardship did not change during the whole study period, so we did not mention it in this study. Antibiotic stewardship in our hospital is mainly performed according to the "Clinical Guidelines for Antimicrobial Drugs" issued by the National Health and Family Planning Commission of the People’s Republic of China in 2015.

Comment 2

The control group age was lower than in the intervention cohort but this was not explained.

Our response:

The average age in the two groups was close, with a difference of 1.73 years (52.14 years in the control group vs 53.87 years in the intervention group). Several studies have reported that age was an independent risk factor for CRKP/CRAB infection. The older the age, the higher the risk of infection. Although the average age in the intervention group was higher than that in the control group in this study, there was still a decreased incidence of MDROs, indicating a relatively robust result.

We have added the details on lines 282-287.

References:

1) Corey Forde, Bryan Stierman, Pilar Ramon-Pardo, et al. Carbapenem-resistant Klebsiella pneumoniae in Barbados: Driving change in practice at the national level. PLoS One. 2017 May 25;12(5):e0176779.

2) Ninghui Guo, Wencheng Xue, Dahai Tang, et al. Risk factors and outcomes of hospitalized patients with blood infections caused by multidrug-resistant Acinetobacter baumannii complex in a hospital of Northern China. Am J Infect Control. 2016 Apr 1;44(4):e37-9.

Comment 3

As respiratory failure was more common was this a result of Covid-19?

Our response:

We checked the data and found that the rehabilitation unit did not admit patients with COVID-19 before November 2022. There was only one patient with respiratory failure in December 2022. Therefore, we consider that the increased proportion of patients with respiratory failure during the intervention period was not significantly associated with the COVID-19 outbreak.

Similar to age, respiratory failure has been reported as an independent risk factor for MDRO infection, which reflects the severity of the patients’ illness. Although the proportion of respiratory failure in the intervention group was higher than that in the control group in this study, there was still a decrease in the infection rate of MDROs, also indicating a relatively robust result.

We have added the details on lines 287-291.

References: Song Yee Kim, Ji Ye Jung, Young Ae Kang, et al. Risk factors for occurrence and 30-day mortality for carbapenem-resistant Acinetobacter baumannii bacteremia in an intensive care unit. J Korean Med Sci. 2012 Aug;27(8):939-47.

Comment 4

The recorded level of hand hygiene compliance was very high and possibly the result of direct observation rather than the actual rate of hand hygiene.

Our response:

We acknowledge the Hawthorne effect, but there was no significant difference in hand hygiene compliance, liquid soap and ABHR consumption between the two phases.

Comment 5

Environmental swabbing would be expected to reflect the current patient flora as most MDRO would not have prolonged survival in the environment (MRSA, VRE and CRAB are exceptions).

Our response:

Yes, the environmental swabbing just reflect the current patient flora. At the time of four rounds of environmental swabbing, the prevalence rates of MDROs were 7.14%, 7.56%, 4.23% and 6.45% respectively.

Comment 6

Reduction of MDRO in rehabilitation is beneficial and allows patients to be transferred more easily. Care provided by family maybe without hand hygiene could have a major effect on results and screening for MDRO was not performed.

Our response:

Family caregivers had poor hand hygiene compliance and could have a major effect on the results. To reduce the influence, we limited the number of family caregivers during the intervention period. The family caregiver rate decreased from 65% in the pre-intervention period to 38.48% in the intervention period. We mentioned this in our discussion. (lines 297-300)

Indeed, we did not screen for MDROs, which is a limitation of this study, and we discussed it on lines 317-319. More research will be carried out in the future based on the limitations of this study.

Comment 7

The findings are not novel and care bundles have been widely examined. The effects of single mode of intervention would be of more interest. More detail on attempts to control antimicrobial prescribing practice and the relationship between patient bacterial flora and that in and away from their immediate environment is needed.

Our response:

Indeed, the effects of single mode of intervention would be of more interest, which will be the direction of further research. Although care bundles have been widely examined, however, there is less evidence on the strategies for MDRO prevention and control in rehabilitation units. The length of stay in the rehabilitation unit is much longer than that of other patients, and some patients are transferred directly from the ICU to the rehabilitation unit due to early rehabilitation. Therefore, rehabilitation patients are high-risk groups for MDRO infection. Moreover, rehabilitation treatment involves sharing equipment and a high frequency of personnel contact, which increases the risk of transmission of MDROs. To assess the effectiveness of multi-model strategies on infections caused by MDROs in the rehabilitation unit, we conducted this study.

Thank you for these valuable suggestions. We have also made modifications carefully, hoping to further guide our research in the future.

Round 2

Reviewer 2 Report

Thank you for making the amendments.